# Slowdown of glacier velocity emerging in the Zanskar Himalayas

Tirthankar Ghosh<sup>1,2</sup>, RAAJ Ramsankaran<sup>1</sup>, Felicity S McCormack<sup>2</sup>, Andrew N Mackintosh<sup>2</sup> Affiliations

- 1 Department of Civil Engineering, Indian Institute of Technology Bombay, Mumbai, India
- 2 School of Earth, Atmosphere and Environment, Monash University, Clayton, VIC, Australia

Correspondence to: RAAJ Ramsankaran (<a href="mailto:ramsankaran@civil.iitb.ac.in">ramsankaran@civil.iitb.ac.in</a>); Tirthankar Ghosh (<a href="mailto:tirthankar.ghosh@monash.edu">tirthankar.ghosh@monash.edu</a>)

Abstract: Trends in glacier surface velocity provide insight into the response of glaciers to climate change as well as local drivers of ice dynamics. The Zanskar Himalayas are heavily glacierised, but retreating glaciers pose a threat to local and regional water security. Remote sensing provides a tool for observing surface velocity over multiple glaciers in a remote and challenging area for field work, providing key observations for tracking changes in this important region. This study provides a comprehensive analysis of long-term (1992-2023) interannual glacier surface velocity and elevation change for 12 selected glaciers in the Zanskar Basin of the Ladakh Himalayas. We show that glaciers have overall experienced deceleration at an average rate of -2.43 m year-1 decade-1 in this region. This reduction in ice velocity corresponds with a mean glacier surface elevation decrease of ~ -0.21 m yr-1 between 2000–2005, increasing to ~ -0.57 m yr-1 by 2015–2020. While glacier mass loss, particularly through thinning, and associated reduction in driving stress was identified as the primary driver of velocity deceleration, glacier-specific characteristics, such as geometry, topography, debris cover and terminus type, also influenced glacier response. For example, lake-terminating glaciers exhibited local increases in ice velocity near their termini. Overall, our results confirm a reduction in glacier health in this region, as glaciers thin and slow down as a consequence of climate warming.

### 1 Introduction

15

Climate change has severely impacted glaciers across the planet (Bolch et al., 2012a; Hugonnet et al., 2021; Immerzeel et al., 2010; Rounce et al., 2023) as glaciers are highly sensitive to climate forcings and thus serve as a significant indicator of climate change (Mackintosh et al., 2017; Oerlemans, 1989). Globally, glacier mass is projected to decline by  $26 \pm 6\%$  to  $41 \pm 11\%$  by the end of the 21st century (relative to 2015), under emission scenarios that correspond to +1.5 °C and 4 °C warming, respectively (Rounce et al., 2023). The Himalayan-Karakoram region, which is often called the Third Pole or the Water Tower of Asia (Immerzeel et al., 2010; Viviroli et al., 2011), hosts one of the largest volumes of glaciated ice outside of Greenland and Antarctica (Wester et al., 2019). Like all other glaciated regions, Himalayan glaciers have also experienced accelerated glacier mass loss over the last few decades (Brun et al., 2017; Shean et al., 2020). They serve as a source of fresh water and play an important role in the global water cycle. For example, the meltwater generated from Himalayan glaciers and snow

50

65

influences the flow of rivers and caters to a population of over a billion people downstream, recharges river-fed aquifers, and contributes to global sea level rise (Azam et al., 2021; Barnett et al., 2005; Bolch, 2017; Bolch et al., 2012b; Immerzeel et al., 2010).

As the glaciers retreat and thin, ice flow velocities can be impacted (Dehecq et al., 2019). Recent studies indicate that many mountain glaciers are experiencing significant deceleration (Dehecq et al., 2019; Wu et al., 2020; Zhou et al., 2021). Glaciers flow due to their weight and gravity, by the processes of sliding at the bed and internal deformation (Bindschadler, 1983; Weertman, 1957). These flow processes are influenced by internal factors such as ice temperature, glacier geometry, bed characteristics and ice-bed interactions, as well as external factors such as air temperature and precipitation (Cuffey and Paterson, 2010; Iken and Bindschadler, 1986). It follows that glacier velocity estimates are a key proxy for understanding the mass balance of a glacier, where few or no direct measurements are available (Millan et al., 2022; World Glacier Monitoring Service (WGMS), 2023).

Traditionally, glacier velocity has been estimated using on-field measurements (Hooke et al., 1989; Stevens et al., 2023; Vincent et al., 2022). For example, DGNSS (Differential Global Navigation Satellite Systems) can be used to track the position of ground stakes on a glacier over time, from which ice surface velocity can be estimated. This method is reliable, but logistically expensive and time-consuming, and is generally limited to accessible areas. Most importantly, this method returns only point-based measurements, which limits both the spatial extent and temporal coverage that is needed to characterise the evolution of a particular glacier system, particularly mountain glaciers, which are remote and difficult to access (Azam et al., 2014; Dematteis et al., 2021; Patel et al., 2022; Sugiyama et al., 2013; Wagnon et al., 2007). By contrast, satellite-based remote sensing methods provide a range of variables with wide spatial coverage at much higher temporal resolution. Such methods can be used to estimate glacier-wide surface velocity at a variety of scales from regional to global, and are efficient and robust (Li et al., 1998; Scherler et al., 2008; Scherler and Strecker, 2012; Satyabala, 2016; Bhushan et al., 2017; Dehecq et al., 2019; Millan et al., 2022).

With the availability of more satellite datasets, many glacier velocity studies have been carried out in the Himalayas, giving some very interesting insights. Glacier-specific and regional studies have revealed heterogeneous patterns in the velocity of the Himalayan glaciers (Bhambri et al., 2011; Dehecq et al., 2019; Garg et al., 2025; Tripathi et al., 2023). Findings from these studies show that glacier velocity varies spatially and temporally, region-wise and within the same glacier, depending on factors such as elevation, slope, size, debris cover fraction, land vs lake terminating, mass budget and other local conditions (Bhushan et al., 2018; Dehecq et al., 2019). Recent studies integrating surface velocity and glacier surface elevation changes in the Himalayas found substantial heterogeneity in their pattern and trends (Bhambri et al., 2023; Garg et al., 2025). Glaciers in the Garhwal Himalayas exhibited significant surface lowering, associated with a reduction in glacier surface velocity, with Gangotri glacier being one exception with an active terminus (Bhambri et al., 2023). Debris-covered glacier in the region was found to have a heterogeneous effect on ice melt. In another study, focused on glaciers in the Chenab basin in the Western Himalaya, researchers reported a significant slowdown by 54% and 20% in Bhut and Warwan sub-basins, respectively (Garg et al., 2025).

95

Ladakh is located between the Himalayas and the Karakoram, often called the Trans-Himalayan region. Due to its location, it has a semi-arid climate, with low precipitation as compared to other Himalayan regions (Archer and Fowler, 2004). Most glaciers are relatively small glaciers ( $

# 2. Study Area

The Zanskar Basin is a high-altitude, cold desert in the Ladakh region of the Western Himalayas (figure 1). The region can be generally classified as a cold-arid climate, as it is in the dry trans-Himalayan region, where the penetration of the Indian Summer Monsoon is very weak. The majority of the precipitation occurs during winter in the form of snowfall (Kamp et al., 2011; Lee et al., 2014).

The glacierised area within the basin is approximately ~1700 km², comprising around ~1755 glaciers (Randolph Glacier Inventory V7; RGI v7.0; RGI Consortium, 2023). Most of the glaciers in this basin are small cirque glaciers (

Figure 1: a) Study area map of Zanskar Basin in the Ladakh Himalaya (black dashed lines showing the boundary), highlighting the glaciers selected for this study (in green) labelled as DDG (Drang Drung Glacier), HG (Hagshu Glacier) and other unnamed glaciers as G3-G12. Neighbouring glaciers around the region are shown in orange and are taken from Randolph Glacier Inventory (RGI) version 7. The inset map shows the sub-regions of High Mountain Asia (HMA) in black outline. b) shows the field photo of Hagshu Glacier, and c) shows the field photo of Drang Drung Glacier taken during October 2023.

125



Table 1: Characteristics of the 12 selected glaciers in our region. We detail the: mean elevation (m), area (km²), length of the glacier (km), mean aspect (degree), mean slope (degree) and whether the glacier is debris covered or not. The debris cover percentage presented here is estimated from the Supraglacial debris cover dataset v1.0 (Scherler et al., 2018), based on linear spectral unmixing-derived fractional debris cover (FDC). All the information is from RGI v7.

| RGI ID (ID)                   | Mean<br>Elevation<br>(m) | Area<br>(km²) | Length<br>(km) | Aspect (degree) | Mean Slope<br>(degree) | Debris Cover |
|-------------------------------|--------------------------|---------------|----------------|-----------------|------------------------|--------------|
| RGI2000-v7.0-G-14-26914       | 5174                     | 68.36         | 24.73          | 3.49            | 13.90                  |              |
| (Drang Drung Glacier)         | 31/4                     | 08.30         | 24.73          | 3.49            | 13.90                  | 12.05        |
| RGI2000-v7.0-G-14-28727       | 4983                     | 63.88         | 17.80          | 54.32           | 16.73                  |              |
| (Hagshu Glacier)              | 4903                     | 03.88         | 17.80          | 34.32           | 10.73                  | 27.83        |
| RGI2000-v7.0-G-14-26878 (G3)  | 4896                     | 48.33         | 18.17          | 8.89            | 15.78                  | 37.20        |
| RGI2000-v7.0-G-14-28745 (G4)  | 5183                     | 18.90         | 11.09          | 27.84           | 14.87                  |              |
| KG12000-V7.0-G-14-28743 (G4)  | 3163                     | 16.90         | 11.09          | 27.04           | 14.67                  | 15.72        |
| RGI2000-v7.0-G-14-28755 (G5)  | 5013                     | 25.60         | 12.48          | 53.72           | 16.95                  | 16.54        |
| RGI2000-v7.0-G-14-27910 (G6)  | 5356                     | 11.01         | 7.59           | 93.97           | 12.44                  | 3.45         |
| RGI2000-v7.0-G-14-27920 (G7)  | 5162                     | 23.01         | 10.35          | 25.95           | 11.79                  | 6.93         |
| RGI2000-v7.0-G-14-29041(G8)   | 5172                     | 18.87         | 9.91           | 35.90           | 15.24                  | 12.02        |
|                               |                          |               |                |                 |                        | 13.82        |
| RGI2000-v7.0-G-14-28003 (G9)  | 5334                     | 6.31          | 5.47           | 27.90           | 15.72                  | 12.78        |
| RGI2000-v7.0-G-14-32718 (G10) | 5394                     | 25.98         | 11.91          | 51.89           | 13.73                  | 10.58        |
| RGI2000-v7.0-G-14-32678 (G11) | 5259                     | 15.59         | 8.98           | 67.34           | 16.85                  | 10.56        |
| RGI2000-v7.0-G-14-27778 (G12) | 5310                     | 3.85          | 4.20           | 323.52          | 17.95                  | 10.28        |

#### 135 3 Data and Methods

#### 3.1 Satellite Data







We selected satellite images from the Landsat series of sensors to generate annual velocity fields from 1992 to 2023, downloaded from the United States Geological Survey (USGS) EarthExplorer: <a href="https://earthexplorer.usgs.gov/">https://earthexplorer.usgs.gov/</a>, last access: 25 February 2025. In this study, we used the panchromatic band (B8) of the Landsat series (7, 8 and 9) and the Green (B2) band from Landsat 5. Previously, various studies have used Landsat images for velocity field estimation (Altena et al., 2019; Bhambri et al., 2023; Das, 2021; Dehecq et al., 2015; Garg et al., 2025; Nanni et al., 2023). 24 pairs of satellite images for inter-annual velocity (Supplementary Table S1) from Landsat 5, 7, 8 and 9 were manually selected based on the quality of the image available (minimum cloud cover with a 20% threshold, minimum snow cover over glaciers) to reduce noise in the results. Satellite images were selected mostly in the months of September and October, as the end of the melting season (the melt season is generally from May-September) is typically followed by minimal snow cover and less cloud cover fraction and is a consistent season that satisfies the image quality conditions mentioned above. The presence of heterogeneous snow cover over two images may reduce or enhance the textural features and induce a mismatch in the correlation algorithm, leading to erroneous results. A previous study reported the horizontal accuracy of the Landsat sensors to be 





was set, resulting in a resultant output of 60m spatial resolution velocity fields. Step sizes smaller than the search window size may introduce redundancy in measurement (Leprince et al., 2007a, b), thereby improving the accuracy of the velocity estimates compared to steps that are of a similar magnitude to the search window (Fahnestock et al., 2016). The outputs from the COSI-Corr algorithm consist of 3 images: N-S (North-South) displacement component, E-W (East-West) displacement component and SNR (Signal to Noise ratio), which assesses the quality of correlation for each pixel.

### 3.2.2 Filtering Process

To minimize the effect of noise, which impacts the accuracy of the correlation outputs, we applied basic filtering to the N-S and E-W surface displacement components. To remove unwanted correlation results, a threshold of 0.9 was set for the Signal-to-Noise (SNR) ratio (Scherler et al., 2008). The Robustness iteration, which is a quality control loop that ensures the derived displacement or velocity vectors are consistent and robust against errors or noise, was set to 2. First, an initial displacement is calculated from image matching, followed by outlier removal. This process is repeated for the number of iterations defined. 2-4 iterations were found to be working well in most cases of glacier velocity estimation (Leprince et al., 2007a). As the three outputs are generated (N-S, E-W displacement and SNR) we used a replace/discard tool in the COSI-Corr algorithm, which removes unwanted pixel values if the velocities defined at these pixels exceeds a threshold. Here, a value of 200 (meters) was set based on existing literature and the maximum glacier velocity in the Western Himalayan region, which is estimated to be approximately ranging from ~50-90 m year-1, with a few exceptions in our case (Bhushan et al., 2018).

Finally, the resultant displacement was calculated using Eq. 1, based on the N-S and E-W velocity components for each image pair, and the final velocity was evaluated using Eq. 2.

$$D_{xy} = \sqrt{(D_{NS})^2 + (D_{EW})^2},\tag{1}$$

$$V = \frac{D_{xy}}{t},\tag{2}$$

Here,  $D_{xy}$  is the horizontal resultant displacement (meters), V is the final surface velocity (m year<sup>-1</sup>), and t is the time (years) between the two image acquisitions.

The final velocity maps are produced at 60m resolution. The same parameter values were maintained throughout all the image pair processing to ensure the velocity outputs are consistent. Finally, we used a 3-pixel \* 3-pixels median filter on the velocity maps to remove any outliers (noise) within the maps and to smooth the velocity outputs without losing their details. Due to poor correlation in some of the pair-wise correlation processes, data gaps exist in the velocity outputs from 2003-2007 and 2011-2012.

### 195 **3.2.3** Uncertainty Estimation

The uncertainty in the velocity field obtained from remote sensing could be influenced by several factors, including satellite scene characteristics (e.g., cloud cover over the region of interest, shadow, etc), surface heterogeneity (difference in snow



cover), image co-registration error, and performance of the feature-tracking algorithm. To minimise the error due to the first two factors, we selected images with a minimal cloud cover fraction 

### 215 3.3 Glacier Surface Elevation Change

We utilised the open-access available global glacier elevation change datasets from (Hugonnet *et al.*, 2021; accessed from https://doi.org/10.6096/13). The dataset is available for the period of 2000-2019, with the mean glacier surface elevation change rate data generated using ASTER (Advanced Spaceborne Thermal Emission and Reflection Radiometer) stereo pairs calculated across four different timeframes: 2000-2005, 2005-2010, 2010-2015 and 2015-2019, at a spatial resolution of 100m.

We analyzed the glacier surface elevation change rate dataset using a widely used method of binning the data into multiple elevation bins (*dh*=50*m*). Bin-wise elevation change analysis is a crucial tool for deciphering the spatial heterogeneity of glacier response to climate forcing, enabling a nuanced understanding of how different altitudinal zones contribute to overall glacier mass balance and dynamics. Unlike whole-glacier averages, which can mask local variability, bin-wise analysis reveals where the most pronounced thinning or thickening occurs, often highlighting elevation-dependent feedback in melt, accumulation, and ice flow. For example, as surface elevation lowers and ice thins, the gravitational driving force decreases, leading to a progressive slowdown of ice flow (Cuffey and Paterson, 2010). There could be some exceptions, for example, differential surface thinning rate, leading to increased surface slope, may increase the driving stress. Mathematically, the driving stress τ(x) is expressed as,





$$\tau(x) = \rho g H(x) \frac{\partial S}{\partial x}(x), \tag{5}$$

In this equation,  $\tau(x)$  is the driving stress,  $\rho$  is the density of ice, g is the acceleration due to gravity, H(x) is ice thickness, and S(x) refers to the ice surface at position x and  $\frac{\partial S}{\partial x}(x)$  represents the surface slope of the glacier at position x along a given flow line.

By contrast, sliding at the bed of the glacier can also play a significant role in ice flow, but sliding is poorly constrained due to a lack of observations. It depends on bed roughness, thermal regime of the glacier and subglacial hydrology (Bindschadler, 1983; Weertman, 1957).

### 3.4 Data analysis and supporting datasets

The central flowlines of the glaciers were manually delineated from high-resolution satellite data for the analysis of velocity profiles. For all other analyses, including glacier elevation changes vs glacier velocity, the datasets were resampled to the spatial resolution of elevation change data (100m), and a reference DEM (Digital Elevation Model) was used to make all the elevation-wise analyses, such as binning the data into different elevation zones. The glacier boundary from RGI Version 7 (Randolph Glacier Inventory V7) was used for the glacier extents and for calculating all the geospatial statistics.

For analysing the climate trend (temperature and precipitation) of the region, we utilised the ERA5-Land Reanalysis climate dataset. ERA5-Land has a spatial resolution of ~9 km, and temporally it provides an hourly dataset from 1950 to the present (Copernicus Climate Change Service, 2019). Data from the nearest grid to the study location were extracted for further analysis.

### 4. Results

# 4.1 Glacier Velocity Trends and Surface Elevation Change

We analysed interannual glacier surface velocity changes across 12 glaciers using the satellite-derived datasets described in section 3, with some data gaps. A spatial map of glacier velocity for the year 2020-2021 is shown in figure 2. Overall, the glacier flow speeds were generally greater in larger glaciers than in smaller ones. We observed a typical velocity distribution, where ice flow increased from the margins toward the centreline and decreased from the accumulation area towards the terminus. In addition, all glaciers demonstrated a very similar pattern of low glacier velocity near the terminus (~<15m year<sup>-1</sup>). In general, the velocity gradually increased with the surface slope, at times by as much as 50% around sudden changes in slope, also coincident with crevassing.

The results reveal temporal and spatial heterogeneities in ice flow. That is, while some glaciers exhibited a marked deceleration over time, others maintained a relatively constant velocity or showed inter-annual accelerations during specific periods. Despite, interannual velocity variations, the median flow speed over the central flowline across all the glaciers varied from 31.11±8.57 m year<sup>-1</sup> in 1992 to 26.25± 1.01 m year<sup>-1</sup> in 2023 (~ -16%), with a minimum median velocity of

24.50±5.73 m year<sup>-1</sup> during 2010, and a maximum of 46.07±5.09 m year<sup>-1</sup> from 2000-2002. The overall region-wide analysis for all the selected glaciers showed a statistically significant decrease in median glacier flow speed over the study period (*p*=0.027, estimated using the t-statistic test; Kim, 2015), at a rate of -2.43 m year<sup>-1</sup> decade<sup>-1</sup> (figure 3). The fastest flowing glacier is the Drang Drung Glacier (DDG) with a median velocity of ~57.98 m year<sup>-1</sup> along the flowline during 1992-2023, followed by G10 with a median velocity of 38.52 m year<sup>-1</sup>. Interestingly, both glaciers are lake-terminating (discussed more in section 5.3). By contrast, the slowest-flowing glacier is G12 with a median velocity of ~9.62 m year<sup>-1</sup>, which could be related to the glacier geometry and ice thickness. While the overall trend of decreasing velocity was statistically significant, trends for individual glaciers were not necessarily statistically significant. In particular, while majority of the glaciers showed statistically significant trends, trends for DDG, G5, and G10 are not statistically significant (figure 3).




Figure 2: Glacier velocity in m year<sup>-1</sup> from 2020-2021 for the selected glaciers (with extents outlined in black), subdivided into three different groups. The left vertical panel (a) shows all the glaciers investigated with insets on the right outlined by yellow boxes; b) velocity for glaciers DDG, HG, G3, G4 and G5; c) velocity for glaciers G6, G7 and G8; d) velocity for glaciers G10 and G11.

Figure 3: a) Median velocity (m year<sup>-1</sup>) for all the glaciers. (b-m) Median velocities (m year<sup>-1</sup>) for individual glaciers from DDG to G12. Each panel shows the interquartile range (shaded orange and blue box) with the median denoted by the black horizontal line, lines and whiskers denote the maximum value (upper) and minimum value (lower), excluding the outliers. The trend is reported in the upper right corner of each panel (which considers the data gap).

We considered three decades from our study period (1992-2000, 2001-2010, 2011-2023) to understand decadal flow pattern changes. Despite the data gaps between 2001-2010, we can discern a broad understanding of changes in the flow trend over this 31-year period. From 1992-2000, all glaciers exhibited an increase in velocity. In contrast, from 2001-2010 and 2011-2023, all velocities decreased except for G5 (which accelerated from 2011-2023) and G11 (which accelerated from 2001-2010). To understand how glacier velocity changed with elevation, we divided the flow trend into two elevation groups: part

of glaciers with elevations less than 5000m (considering the mean elevations for the glaciers) and the rest of the glaciers' part which lie higher than 5000m. The median velocities within both elevation groups showed a declining trend at rates of -3.46 m year<sup>-1</sup> decade<sup>-1</sup> and -1.19 m year<sup>-1</sup> decade<sup>-1</sup>, respectively. This suggests the glacier velocity declined faster near the ablation zone (

Figure 4: Box plots for the median velocity (m year-1) for the region of glaciers with elevation a) below 5000m; and b) above 5000m. In each panel, the median velocity trend is denoted by the black horizontal line, the interquartile range is shown in the coloured box, and the lines and whiskers denote the maximum value (upper) and minimum value (lower), excluding the outliers. Outliers are not included in the plot.

Velocity profiles along the central flowline are shown in figure 5. The patterns correspond with glacier hypsometry. We used the Hypsometric Index to calculate and classify the glaciers as very bottom-heavy (HI > 1.5), bottom-heavy geometries (1.2 < HI  $\leq$  1.5), very top-heavy (HI < -1.5), top-heavy (-1.5 < HI < -1.2), and equidimensional (-1.2 < HI < 1.2) (Jiskoot et al., 2009). Glaciers like G4, G8, and G11 exhibited a bottom-heavy geometry, while HG and G3 are very bottom-heavy (HI > 1.5), with a larger share of area at lower elevations. Such geometries can enhance driving stress in the ablation zone due to thinning-induced surface steepening and increased ice mass concentrated at lower elevations (Bhushan et al., 2018; Sam et al., 2018).


290

295

Table 2: Overall median velocity trend (m year<sup>-1</sup>) for individual glaciers over the period 1992-2023. Uncertainty is calculated as one standard deviation (m year<sup>-1</sup>) and the *p*-value denotes a statistically significant trend, as calculated using the t statistic-test.

| Glacier      | Trend (m year-1 decade-1) | Uncertainty<br>(1σ) | <i>p</i> -value |  |
|--------------|---------------------------|---------------------|-----------------|--|
| DDG          | 0.59                      | 1.25                | 0.63            |  |
| HG           | -4.12                     | 1.39                | 0.00            |  |
| G3           | -2.96                     | 1.29                | 0.03            |  |
| G4           | -2.78                     | 0.97                | 0.01            |  |
| G5           | -3.74                     | 1.85                | 0.05            |  |
| G6           | -2.11                     | 0.74                | 0.01            |  |
| G7           | -5.32                     | 1.88                | 0.01            |  |
| G8           | -3.34                     | 1.09                | 0.00            |  |
| G9           | -3.89                     | 0.86                | 0.00            |  |
| G10          | 1.75                      | 1.89                | 0.36            |  |
| G11          | -6.40                     | 1.36                | 0.00            |  |
| G12          | -1.66                     | 0.42                | 0.00            |  |
| All Glaciers | -2.43                     | 0.87                | 0.00            |  |

Figure 5: (a-l) Scatter plots showing the glacier velocity profile along the central flowline for all the glaciers from 1992-2023. The x-axis represents the distance along the flowline (km) from the start of the terminus (marked by the origin). The different colours represent data points from different years. The black thick line refers to the mean velocity across the time period.

Glacier surface elevation change revealed a clear altitudinal dependency, with consistent thinning patterns within each elevation bin over the period 2000–2020. Across the 12 glaciers analyzed, thinning generally increased in the lower elevation bins, particularly below ~4700 m, indicating stronger ablation and dynamic thinning towards glacier termini (figures 6 and 7). This pattern persisted across the period (2000-2020), with the lowest elevation bands consistently exhibiting the greatest thinning rate (*dh/dt*) of ~ -1.3 m year<sup>-1</sup>. While high-elevation regions (>5000 m) also experienced surface lowering, the rates were relatively lower, likely reflecting reduced melt and less dynamic thinning at those elevations.

The overall median glacier surface elevation change rates for all the glaciers in the study region show a clear acceleration of thinning, from -0.22 m year<sup>-1</sup> (2000–2005) to -0.57 m year<sup>-1</sup> (2015–2020). This pattern aligns with glacier hypsometry theory, as bottom-heavy glaciers (e.g., HG, G3, G4, G8, G11) experienced greater thinning, with HG showing especially strong losses



in the most recent period (-0.72 m year<sup>-1</sup>). The largest glacier DDG, an equidimensional glacier (where glacier area is more evenly distributed across its elevation range), experienced a median glacier-wide thinning rate of – 0.06 m year<sup>-1</sup> during 2000-2005, to – 0.63 m year<sup>-1</sup> during 2015-2020. Few glaciers (HG, G3 and G6) showed a median positive rate during 2000-2005, which could be linked to the higher accumulation rate in higher elevations (figure 7). Supplementary section (Table S3) provides detailed glacier elevation bin-wise change rates.

Figure 6: (a-l) Glacier hypsometric curve of surface elevation change. The left y-axis shows glacier area (km²), represented as grey histograms and the right y-axis shows the glacier surface elevation change rate, dh/dt (m year-¹) from Hugonnet et al, 2021, data for four different periods, 2000-2005, 2005-2010, and 2015-2019 represented in red, blue, green and purple lines, respectively. The colored bands represent the  $1\sigma$  (standard deviation).

Figure 7: Glacier-wide mean elevation change (dh/dt) in m year-1 for different periods. Blue denotes thickening and red denotes thinning.




Figure 8: Glacier surface elevation change (Hugonnet et al, 2021). a-b: show the elevation change map for different periods (2000-2005 & 2015-2020). The colour bar represents the 2<sup>nd</sup> -98<sup>th</sup> percentile of elevation change. The inset in panel d shows the tongue of the Drang Drung glacier and its neighbouring glaciers.

## 4.3 Elevation-wise Glacier Velocity Trend



Elevation bin-wise velocity analysis also revealed inter-glacier variability associated with glacier geometry changes and surface elevation lowering. We evaluated elevation bin-wise (bin size=50m) velocity evolution across different periods (1999-2000, 2009-2010, 2014-2015 and 2019-2020), finding velocity trends that coincided with specific elevation bands, which sheds light on the underlying dynamics influencing the trends. We found that most glaciers in our study exhibited similar patterns of glacier slowdown across the glacier, with the greatest reductions occurring near the ablation zone (Figure S1). For instance, in DDG (panel b), the peak velocity during the 2000–2005 period exceeds 40 m year<sup>-1</sup>, while in the most recent period (2015–2020), velocities have significantly reduced across the entire elevation range. In other cases (e.g., panels c, g, l), the velocity profiles exhibited a marked shift in both the peak magnitude and elevation position, suggesting changes in flow dynamics, which could be potentially driven by mass loss and evolving ice geometry. Some glaciers, such as G9 (panel f) and HG (panel h), exhibited more complex or less consistent trends, possibly reflecting the interplay of local topography, debris

cover, and variations in surface elevation change. Several glaciers, such as HG, G4, and G5, showed velocity increases at their lower elevations in 2019-2020.

### 4.4 Non-climatic factors

We used a correlation matrix to understand the interplay between glacier surface velocity and different non-climatic factors such as mean elevation, area, mean slope, mean aspect, and length (Figure 9). From the matrix, it can be said that glacier length and area have the strongest positive correlations with median velocity, with a correlation coefficient of 0.75 and 0.67, respectively. In contrast, slope (r = -0.37) and aspect (r = -0.36) shows a moderate negative correlation with velocity. The mean elevation shows almost no correlation (r = 0.01) with surface velocity.

Figure 9: Correlation matrix between median surface velocity and non-climatic variables such as Mean elevation (m), Area (km²), Slope (in degrees), Aspect (in degrees) and length (km)




### 4.4 Climatic trends

We analyzed climate data over the region from the ERA5-Land dataset, calculating trends over the period 1990-2025. The overall mean annual temperature increased by  $0.02\pm0.01$  deg C year<sup>-1</sup> over the period, with increases in both summer (May-September;  $0.01\pm0.01$  deg C year<sup>-1</sup>) and winter (October-April;  $0.03\pm0.01$  deg C year<sup>-1</sup>) temperatures (figure 10a). Summer precipitation (MJJAS) remains consistent with no significant change. In contrast, the winter precipitation shows a declining trend of -0.0003mm year<sup>-1</sup> (figure 10b).

Figure 10 a) Trend of summer temperature (MJJAS), winter temperature (ONDJFMA), and annual mean temperature from ERA5 Land; b) Summer precipitation and winter precipitation (1990-2025) from ERA5 Land. Each data point for temperature is the mean, whereas for precipitation, it is cumulative.

### 5. Discussion







# 5.1 Trends in Surface Velocity and Elevation Change

Our analysis shows an overall consistent slowdown in glacier velocity over recent decades. This trend aligns well with other regional and glacier-specific observations of glacier deceleration, which are linked to ongoing thinning and mass loss. For instance, Dehecq et al. (2019) reported a significant decadal slowdown (2000-2017) in glacier flow across High Mountain Asia. Especially, the Lahaul-Spiti region, within our study area, exhibits an average velocity trend of  $-4.6 \pm 0.6$  m year<sup>-1</sup> decade<sup>-1</sup>. Similar patterns have been reported for Parkachik Glacier in the Suru Basin, adjacent to our study area, where velocity decreased from  $45.18 \pm 1.78$  m year<sup>-1</sup> in 1999-2000 to  $32.28 \pm 0.80$  m year<sup>-1</sup> in 2020-2021, primarily linked to mass wastage, debris-cover increase, and reduced accumulation (Rana et al., 2023). Another study calculated the long-term change (1993-2018) of Parkachik Glacier, showing evidence of velocity slowdown by  $\sim 35\%$  (10.58  $\pm$  5.68 m year<sup>-1</sup>) due to increased debris cover near the terminus and mass loss (Garg et al., 2022b). A recent study focusing on the Chenab Basin showed glaciers slowing down by 54 % and 20 % between 1993-2021 in the Bhut and Warwan subbasins, respectively, which was primarily linked to increased debris-cover over glaciers. Another recent study by (Bhattacharjee et al., 2025) reported a velocity of 0.26  $\pm$  0.02 m day<sup>-1</sup>, which corresponds to 94.97  $\pm$  7.31 m year<sup>-1</sup> for DDG close to the ELA (Equilibrium Line Altitude) and 0.04  $\pm$ 0.003 m day<sup>-1</sup>, equivalent to 14.61 ± 1.10 m year<sup>-1</sup> for G3 near the terminus, which aligns very well with our results. Although previous studies have reported a slowdown of glaciers in the Ladakh region, a contrasting increase in glacier velocity was found in the Drang Drung Glacier. For example, the results from the study by (Singh Jasrotia et al., 2024) show velocity increased from  $71 \pm 6.1$  m yr<sup>-1</sup> in 1999-2000 to  $140 \pm 7.4$  m yr<sup>-1</sup> in 2019-2020 (by  $\sim 50$  %), which contrasts with our findings, and is likely an artefact of their consideration of two timeframes only. Overall, our declining glacier velocity trends align with patterns observed for nearby glaciers in other studies.

The slowdown observed could be primarily explained by sustained glacier thinning, resulting in reduced driving stress, especially at lower elevations in the ablation zones, often exceeding several meters per year in some cases. Detailed investigations of surface elevation change revealed strong glacier thinning across all the glaciers, with higher rates observed from 2010-2020. HG and G8 exhibited the highest negative elevation change (0.72 m year<sup>-1</sup>) for 2015-2020, despite them being debris-covered glaciers. This pattern matches with results reported by Bhushan et al., (2018), where debris-covered glaciers exhibited similar elevation change compared to clean ice glaciers. This could be likely due to melt-enhancing features on debris-covered ice, such as ice cliffs and supraglacial ponds that conduct a significant amount of surrounding heat into the ice, resulting in ablation (Brun et al., 2016; Pellicciotti et al., 2015; Reznichenko et al., 2010). A geodetic mass balance estimated by (Mandal et al., 2024b) also reports an elevation change rate of  $-0.44 \pm 0.09$  m a<sup>-1</sup> (2000-2017) for Western Ladakh glaciers that covers Zanskar Basin as well, which aligns well with Hugonnet et al., 2021 trends.

Several glaciers in our domain (DDG, HG, G4 and G5) showed higher velocities in lower elevations in recent years, accompanied by lower thinning rates near those elevations (figure 6). Among them, G4 and G5 are top-heavy glaciers, while DDG has a similarly sized accumulation and ablation zone, likely maintaining higher ice flux towards lower elevations. In






contrast, HG is a heavily debris-covered glacier with a bottom-heavy hypsometry, which is likely more sensitive to ablation zone dynamics. Increased velocity near this zone could be potentially linked to melt-induced basal sliding. Bhushan et al. (2018) also reported a dynamically active trunk of DDG (>20 m year<sup>-1</sup> in 2013–2014), consistent with our median flowline velocities. This highlights the complex interplay between debris, melt, and dynamics.

Overall, our findings reinforce the broader picture of sustained glacier slowdown across the Zanskar region, which is closely tied to glacier mass loss due to thinning, and is consistent with the patterns observed in other studies and existing theory. The climate data analysis further supports this link, showing increasing summer temperatures and declining winter precipitation (snowfall) over recent decades (figure 10), which is likely to intensify in the future under continued warming.

### 5.3 Velocity trend of lake-terminating glaciers

Glacier velocities across the study region show clear differences between lake-terminating and land-terminating glaciers, with the former generally maintaining higher surface flow speeds – often up to twice that of their land-terminating counterpart. For instance, median velocity along the central flowline for DDG and G10 corresponds to 57.98 m year<sup>-1</sup> and 38.52 m year<sup>-1</sup>, respectively, which is significantly higher than other glaciers.

The presence of a pro-glacial lake has been previously linked with enhanced glacier mass loss (King et al., 2019; Minowa et al., 2023; Zhang et al., 2023). That is, a lake-terminating glacier loses mass at its terminus by surface melt and frontal ablation, which includes mechanical calving and subaqueous melt (Carrivick and Tweed, 2013), both of which drive dynamic thinning and retreat. This is evident from the elevation change analysis of both DDG and G10 (figure 6). For DDG, retreat rates of 21.11 m year<sup>-1</sup>, and a total retreat of 925m have been observed between 1971 and 2017, with the pro-glacial lake developing around 2014 (Rashid and Majeed, 2018). This dynamic thinning and retreat could be related to calving, which was evidenced in 2023 (e.g. see supplementary fig S1). The pro-glacial lake near the G10 terminus was formed well before 1975 (Govindha Raj, 2010). In contrast to DDG, calving at G10 is not apparent from satellite observations, and it is less clear what mechanisms underlie the thinning of this glacier. Such lake expansion could alter the local force balance near the termini, promoting frontal acceleration (Pronk *et al.*, 2021).

Numerical modelling experiment on an alpine glacier has also explained that the presence of a pro-glacial lake enhances the glacier retreat by 4 times and induces ice acceleration by 8 times higher as compared to land-terminating glaciers forced with the same climatic parameters (Sutherland et al., 2020). The contrasting behaviour of lake-terminating glaciers could therefore be explained in two ways: (a) less buttressing and altered local force balance at the ice-lake margin, and (b) enhanced surface lowering near the terminus. The latter, in particular, influences the flow characteristics by regulating the glacier's dynamic thinning phenomenon, as also discussed and found in the majority of lake-terminating Himalayan glaciers (King et al., 2019; Pronk et al., 2021).

Figure 11: Velocity near the terminus of the lake terminating glaciers. a) DDG; and b) G10 for the year 2020-2021. Highlighting the high velocity near the terminus for G10 and DDG. 'DC' represents the Debris Cover area along the glacier tongue. The basemap is a Landsat satellite image from September 2021.

#### **5.4 Limitations**

While the study provides some critical insights into the evolution and control of interannual glacier velocity for selected glaciers in the Zanskar region of Ladakh, a few limitations need to be acknowledged. First, issues in velocity retrieval led to data gaps between 2002 and 2008, primarily due to the limited availability or poor quality of optical satellite imagery. Persistent cloud cover and low-contrast surface conditions further hindered reliable feature tracking and image correlation over the given time frames. These data gaps in time series analysis may miss specific events during the acceleration or slowdown phase, especially during periods of high inter-annual variability, such as high-melting years, which can lead to biases in the trend estimates. Second, while we analysed the interannual to interdecadal velocity changes in relation to surface elevation change and other non-climatic factors, short-term or seasonal velocity trends – e.g. driven by meltwater input to the ice-bed interface and subglacial conditions – were not resolved in this study. Finally, erroneous velocity due to the presence of cloud cover, shadows, or snow may occur. This may induce random errors, resulting in localized zones of high uncertainty which require identification and correction (Troilo et al., 2024). The estimated uncertainties in our results are well within limits and less than the magnitudes of velocity observed in the terminus of DDG and G10.

### 6. Conclusions


505

We set out to understand the long-term evolution of glacier velocity and its potential drivers in the Zanskar Basin of the Ladakh Himalayas. By analyzing interannual glacier velocity changes from 1992 to 2023 alongside glacier surface elevation change trends over four distinct periods between 2000 and 2019, of 12 selected glaciers, we provide a comprehensive picture of how glacier dynamics have evolved over time.

The results highlight an overall trend of glacier slowdown across all the glaciers, accompanied by prolonged glacier thinning, highlighting the dynamic response of the glaciers to sustained mass loss. Surface thinning has been more pronounced at lower elevations (

Author contributions. TG- Conceptualisation, methodology, formal analysis, Writing- original draft, reviewing and editing, RR- Conceptualisation, Writing- reviewing and editing. FM-Writing- reviewing and editing, AM- Writing- reviewing and editing. RR, FM and AM jointly supervised the study.

Competing interests. FM is a member of the editorial board of *The Cryosphere*. All other authors declare that they have no conflict of interest.

Supplements. Supplementary Table S1: Details of Satellite images and DEM used for annual velocity estimation and analysis, with sensor name and spatial resolution, Supplementary Table S2. Details for satellite image pairs used for glacier surface velocity, with the estimated uncertainty, Supplementary Table S3: Detailed elevation bin-wise glacier surface elevation changes statistics for each glacier. The elevation data is binned into 50m bins for all the data analysis. Supplementary Figure 1: Velocity distribution of different glaciers in different elevation zones. Supplementary Figure S2: field photographs show a)
 the terminus of DDG in contact with the glacial lake; b) the presence of icebergs (marked in yellow box) floating in the glacial lake, indicating that DDG undergoes mechanical calving events. Photographs are from October 2023.

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
