# Peer review of "Slowdown of glacier velocity emerging in the Zanskar Himalayas"

_EGUsphere, 2025_

## Referee Comment (RC2)

**Review of – Slowdown of glacier velocity emerging in the Zanskar Himalaya**

By Tirthankar Ghosh, RAAJ Ramsankaran, Felicity S McCormack, Andrew N Mackintosh

In this manuscript, the authors quantify multi-decadal change in glacier dynamics in the Zanskar Basin (Ladakh Himalaya) by estimating interannual surface velocities from Landsat feature tracking (1992–2023) and relating these to surface elevation change from the Hugonnet et al. dataset. The authors analyse 12 selected glaciers spanning a range of sizes, slopes, aspects, debris cover fractions, and terminus types. The manuscript is well written overall and complements other available papers on the region. However, some revisions would be useful to improve the final version of this. Firstly, given that most of the results in this paper are based on the optical feature tracking velocity maps, it would be ideal to either make these rasters available directly or at a minimum add some vector plots to the supplementary material. The absence of this makes evaluating underlying data quality more challenging. The discussion of the processes driving glacier acceleration and slowdown could use some edits, particularly capturing the more complex links to glacial hydrology that are currently absent. Also, the discussion of non-climatic drivers (including lake impacts) has some weaknesses in the causality inferences, with differences in glacier size/thickness not being discussed. None of these revisions will fundamentally alter the manuscript and I expect it to be suitable for publication in TC with appropriate edits.

Line by line comments:

The title could be clearer – there's some ambiguity around what 'emerging' means

L13 – a brief word about why they were selected – 'representative'? 'distrubuted'? 'large' ? might help

L20 'glacier health' avoid this term. Spell out what you mean. Also you just noted acceleration of some lake termini – would this not be an acceleration in the face of climate change? Maybe caveat to capture the complexity.

L25 Perhaps note here these figures exclude the ice sheets. If we include Antarctica/Greenland they are much lower.

L28 glaciated ice -> ice

L32 'caters to a population of over a billion people' while >1 billion live in the catchments, the glacial water fraction is negligible for many of these. As written this somewhat oversells the importance of glacial water (which is nevertheless locally crucial). Please reword to better capture this.

L35-40 Glacier basal hydrology is of course a critical driver of velocity which can substantially complicate this relationship. Many glaciers undergo a large seasonal cycle in velocities which of course does not reflect equivalent seasonal mass changes. Can you please say a little more about this here, and how it can be mitigated when interpreting changes in velocity (e.g. long timeseries, multi-glacier analyses, context-aware). You do this all already so this can strengthen your case.

L43-45 Worth noting here that people have been doing satellite-based feature displacement tracking for ice velocity basically as long as GNSS has been around.

L50 Satellites do not provide higher temporal resolution than a ground-based GNSS station. You can get sub-minute timings from the latter. This paper may be of interest to review – we compare satellite and ground based data (https://doi.org/10.5194/tc-17-4063-2023).

L55 Agreed that they are interesting but perhaps can be worded differently here. Agarwal et al might be relevant for this section too – a detailed regional analysis with some similar objectives to your study https://doi.org/10.1016/j.scitotenv.2023.165598.

L70-75 Could you speak on the concept of peak water here? This matters more for water security than the 'glacier mass loss' values – if peak water is still in the future then greater mass loss might actually lead to a short term increase in water availability. The timescale of interest matters here. Huss and Hock have a global compilation if there are no specific local studies.

L86 These are some good studies overall, and you might rewrite this sentence in a more positive way e.g.'despite the insight these provide into xx, gaps in our understanding of yy remain'.

L87 Does Dehecq et al not cover the whole HMA including this area?

L92 'glacier flow trend' -> 'glacier surface velocity trend'

L94 What 'reanalysis' was done here?

L96 This seems a risky aim as worded, as exact determination of these processes is not always possible, particularly with only this remote sensing data.

L97 This is not clear. Do you mean whether temperature vs precipitation are the key drivers? Can you expand this question to make it clearer.

L139 Can you say exactly which Landsat mission you got images from. Were you forced to use post-SLC failure L7?

L153 Did you just do sequential image matching across years (e.g. 2015-2016)? Or was multiyear matching also done (e.g. 2015-2017, 2018, 2019). For these v. slow glaciers with limited decorrelation the latter can be useful and reduces dependence on single images but unsure what was done here.

L171 Could you briefly note how the SNR is calculated or reference a paper that does. This is calculated in a few different ways.

L191 – 194 If using a median smoothing kernel then "smooth the velocity outputs without losing their details" is probably not true, rather "smooth with acceptable loss of detail". Worth noting that this procedure commonly erodes the glacier boundary by 1-2 pixels with the 'stationary' pixels winning out on the median. Probably still acceptable but worth noting especially for small glaciers.

L220-225 This is described as a more information rich-approach than 'whole glacier averages' – but the baseline Hugonnet dataset is already a spatially distributed dataset so this is losing information. Might be worth instead framing it in terms of noise suppression and interpretability (i.e. dimensionality reduction).

L225-226 this is only true for 'all else equal' – as your subsequent lake analyses show is not always true.

L247 onwards – I am not sure that the velocities reported to the nearest 0.01m are warranted here.

L284 Looking at the graphs for G5 and G11 that doesn't seem to tell the whole story.

L289 Is 5000m exactly the ELA on all these glaciers? If not, can you either repeat this with the actual ELA or reword?

L315 Could you plot the annual velocities as lines rather than scatterplot? It is very hard to keep track of. If the lines are colored with a continuous gradient from start-end it becomes easier.

L375 To me this reads as all covariates of ice thickness – area and length are themselves strongly correlated (0.97) and both will relate to how large and thick the ice is. Slope anticorrelating with velocity is contrary to usual expectations, but should also be because slope and thickness anticorrelated.

L426-432 This would be a good place to say more about the subglacial hydrology links to velocity which is currently missing.

L439-441 In isolation I am not sure how much this tells us – they could simply be larger/thicker glaciers as this has not been controlled for.

L447 Not sure you have discussed/shown evidence for dynamic thinning here.

L460 The high velocity is not particularly apparent in these figures. Here and throughout it would be good to add the velocity vectors on top of the speed field, this is helpful for evaluating data quality. Also, could you use colour base images – the lakes would be much clearer.

L480 perhaps 'detailed' rather than 'comprehensive'

L485-487 I am not sure what this sentence is getting at. As I mentioned above, I think most of this can be summed up as 'larger glaciers tend to be thicker and flow faster' – not particularly insightful in itself.

L487-488 Again, the absolute velocity being different does not tell us much unless you have controlled for some other parameters here. If we compare a large lake terminating glacier with a small, thin land terminating glacier we cannot attributed the absolute velocity difference only to the lake.

-Max